# The Mediation of miR-34a/miR-449c for Immune Cytokines in Acute Cold/Heat-Stressed Broiler Chicken

**DOI:** 10.3390/ani10112168

**Published:** 2020-11-20

**Authors:** Tao Li, Yiping Song, Xiuyu Bao, Jianqin Zhang

**Affiliations:** College of Animal Science and Technology, Northwest Agriculture and Forestry University, Xianyang 712100, China; 18392459744@nwafu.edu.cn (T.L.); 18735431454@nwafu.edu.cn (Y.S.); xiuyu@nwafu.edu.cn (X.B.)

**Keywords:** miR-34a, miR-449c, heat stress, immune cytokines

## Abstract

**Simple Summary:**

In the intensive and scale poultry industry, the level of heat stress (HS) directly affects the growth, development, and production performance of poultry. To alleviate the adverse effects of stress in broilers, microRNA (miRNA) was regarded as a potential regulator of immune cytokines. In this study, through the sequencing analysis of spleens after cold/heat stress, we found that 33 and 37 miRNA were differentially expressed in the heat stress group compared with the normal (NS) group and cold stress (CS) group, respectively. The differential miRNA were mainly involved in biological processes such as the cytokine–cytokine receptor interaction. To further understand the miRNA-mediated effect of heat stress on the immune level of chickens, we selected miR-34a and miR-449c as the research objects, predicted and verified that interleukin 2 (IL-2) and interleukin 12α (IL-12α) were the target genes of miR-34a and miR-449c. Coupled with the analysis of the expression of other cytokines, we found that miRNA could change the expression of immune cytokines directly or indirectly. This discovery provides a new insight into the mediation of miRNA for immune cytokines in acute cold/heat stressed broiler chicken.

**Abstract:**

An increasing amount of evidence has revealed that microRNAs (miRNAs) participated in immune regulation and reaction to acute cold and heat stresses. As a new type of post-transcriptional regulatory factor, miRNA has received widespread attention; However, the specific mechanism used for this regulation still needs to be determined. In this study, thirty broilers at the same growth period were divided into three groups and treated with different temperature and humidity of CS (10–15 °C and 90% Relative Humidity (RH)), HS (39 °C and 90% RH), and NS (26 °C and 50–60% RH) respectively. After 6 h, splenic tissues were collected from all study groups. miRNA sequencing was performed to identify the differentially expressed miRNAs (DEMs) between HS, CS, and NS. We found 33, 37, and 7 DEMs in the HS-NS, HS-CS, CS-NS group. Gene ontology (GO) and Kyoto Encyclopedia of Genes and Genomes (KEGG) pathway analysis showed that DEMs were significantly enriched in cytokine–cytokine receptor interaction and functioned as the cellular responders to stress. We chose two miRNA, miR-34a and miR-449c, from the same family and differential expressed in HS-CS and HS-NS group, as the research objects to predict and verify the target genes. The dual-luciferase reporter assay and quantitative real-time PCR (qRT-PCR) confirmed that two cytokines, IL-2 and IL-12α, were the direct target genes of miR-34a and miR-449c. To further understand the mediation mechanism of miRNAs in acute cold/heat-stressed broiler chicken, a splenic cytokines profile was constructed. The results showed that IL-1β was strongly related to acute heat stress in broiler chicken, and from this we predicted that the increased expression of IL-1β might promote the expression of miR-34a, inducing the upregulation of interferon-γ (INF-γ) and IL-17. Our finds have laid a theoretical foundation for the breeding of poultry resistance and alleviation of the adverse effects of stress.

## 1. Introduction

Heat/cold stress is defined as a condition that occurs when an animal cannot dissipate or absorb enough heat to maintain a balanced body temperature [1]. In China, whether in livestock or poultry, heat stress is a hindering factor restricting the development of the breeding industry [2,3]. The abnormal phenomenon has resulted in: damage to the internal environment homeostasis, reduction of the food consumption [4], decrease in body weight gain [5], reduction of reproductive [6] and laying performance [7], and depression of the muscle mass and yield [8]. Coupled with the intensive management model, poultry is particularly sensitive to cold/heat stress because of their special physiological characteristics, such as no sweat gland. Heat stress negatively affects the immune responses, growth performance, productivity of broilers and laying hens [9,10]. Cold stress of 12 °C lower than the ambient temperature results in oxidative stress, ultrastructural cardiac damage, immune dysregulation, and inflammation [11]. Most importantly, it would induce a decrease in immunity [12], exposing the birds to the effects of pathogens such as salmonella [13] and virulent infectious bursal virus. It also increases oxidative stress in cells, causing abnormal gene expression and protein synthesis, disruption of DNA or organelle function, and the ultimate death of the cell [14]. The high morbidity and mortality rates in the chickens because of these factors have led to large economic losses in the poultry industry [2]. Therefore, it is pivotal to the industry that a solution is found to reduce the acute heat/cold stresses endured by these animals.

A number of studies have shown that immune function is suppressed directly or indirectly when exposed to stressors [15]. Yadav et al. determined out that heat stress had a negative effect on nutrition absorption by reducing dry matter intake, thus affecting the immune system and inflammatory response [16]. The production of cortisol stimulates the immune system during acute stress, but cortisol secretion is related to immunosuppression during chronic stress [17,18]. As a result, the suppression of the immune system makes animals more vulnerable to disease and immune challenges.

MicroRNA (miRNA) refers to a small (18–25 nt) non-coding RNA which modulates gene expression at the post-transcriptional level [19] and plays an important role in animal and plant growth and development [20,21,22]. It is involved in almost every important physiological and biochemical process [23], such as cell proliferation, differentiation, and apoptosis [24], and is a regulator of heat shock response in mammals. Increasing evidence has revealed that heat stress can change the immune level of the body by affecting the expression of miRNA [25]. Li et al. [26] studied the expression of miRNAs in the mammary glands of dairy cows under heat stress and identified 483 known bovine miRNAs and 139 new miRNAs. Zhu et al. [27] found that 16 miRNAs and 502 genes in the plasma of laying hens changed significantly during heat stress. Some studies have also found that heat stress affects immunity through miRNA. In 2014, 52 miRNAs that had changed significantly due to heat stress, were found in the serum of Holstein cows, and these miRNAs regulated stress and immune response genes [28]. Overexpression of gga-miR-1306-5p can up-regulate the release of pro-inflammatory mediators, including nuclear factor kappa B (NF-kβ), tumor necrosis factor-α (TNF-α), interleukin (IL-6), and IL-1β, resulting in an effect similar to Tollip silencing [29].

T cell proliferation is critical for immune responses; however, T cells demonstrated substantial cell-cycling defects when miR-142 was deficient [30]. Inhibition of miR-34a improved autoimmune arthritis progression accompanied by a downregulated both the number of peripheral T lymphocytes and the expression of cytokine [31]. MiR-15b plays an important role in the immune response to myasthenia gravis disease by directly targeting IL-15 [32]. In terms of cold/heat stress, miR-34a and miR-142-5p were extremely sensitive to temperature. With the rise of body temperature, increased miR-142–5p and miR-143 would target endogenous pyrogens, including IL-6, Interleukin-6 receptor subunit beta (IL6ST), and Tumor Necrosis Factor (TNF) to complete negative feedback [33]. Moreover, miR-34a showed sensitivity to hypothermic treatment following traumatic brain injury (TBI), and the overexpression of miR-34a can increase cellular stress and vulnerability [34]. Rno-miR-34a and Rno-miR-34b were both significantly up-regulated in rat small intestine after heat treatment, suggesting that miR-34 may also be involved in heat stress-induced apoptosis of the intestinal epithelium [35]; regrettably, the concrete mechanism by which this occurs is still unclear.

The aim of this study is to explore the regulatory mechanism of miRNA on the immune level of chickens under cold and heat stress. miRNA sequencing was used to detect the expression of different miRNAs. Meanwhile, we attempted to identify some target genes related to the immune system, thereby revealing miRNAs that exert influence on immunity and stress. Our results will further provide the theoretical basis for poultry anti-stress breeding.

## 2. Materials and Methods

### 2.1. Stress Treatment and Sample Collection

The experimental chickens were purchased from a local henhouse, where they had been raised in the same growth environment, and to the same growth stage, and nutritional status. The entire experimental process and animal treatments were reviewed and approved by the Ethical Committee of Experimental Animal Center of Northwest Agriculture and Forestry University (ethical code: # 0821/2016). A total of 30 Rose 308 broiler chickens of 28-day-old were selected randomly and divided into three groups (HS group, CS group, NS group). Then, 10 broiler chickens in the HS were treated in a hot and humid situation at 39 °C and 90% RH, 10 broiler chickens of the CS group were placed in a cold and humid environment at 10~15 °C and 90% RH., and 10 NS group broiler chickens were maintained at 26 °C and 50~60% RH. After the 30 chickens were maintained in their respective environments for 6 h [36,37], they were slaughtered and their spleen samples were collected. The tissue was washed using diethyl pyrocarbonate (DEPC) treated water (ThermoFisher Scientific, Waltham, MA, USA) and frozen in liquid nitrogen, respectively. Following this, they were stored at −80 °C until the next use.

### 2.2. Small RNA Libraries Construction

Three spleen samples were selected randomly from each group for RNA isolation. Total RNA of spleen samples were isolated separately using Trizol reagent (Takara Bio Inc. Otsu, Shiga, Japan) according to the manufacturer’s instructions. The quality of RNA isolation was checked with Nanodrop 2000 (Thermo Fisher Scientific, Waltham, MA, USA) and the RNA integrity was detected using agarose gel electrophoresis. Subsequently, Small RNA fragments of 18–30 nt were separated and recovered by denaturing polyacrylamide gel electrophoresis (PAGE). A 5′ adaptor (5′-pUCGUAUGCC GUCUUCUGCUUGidT-3′) and a 3′ adaptor (5′-GUUCAGAGUUCUACAGUCCGA CGAUC-3′) were then ligated with small RNA fragments [22]. The linked products were reverse-transcribed to complementary DNA (cDNA), and then PCR amplified and purified on page gel. The constructed library uses Agilent 2100 (Agilent Technologies Inc. Palo Alto, CA, USA) Bioanalyzer and ABI Step One Plus RT PCR System to detect the quality and yield.

### 2.3. miRNA Sequencing and Expression Analysis

Small cDNA libraries were sequenced at the sequencing company (Shenzhen Hengchuang Gene Technology Co., Ltd., Shen Zhen, Guang Dong, China) using Illumina HiSeqTM 2000 system (Illumina, San Diego, CA, USA). The high-quality clean tags obtained by the sequencing platform after data quality controlled (QC) and filtering of raw reads. To search for known miRNAs, the clean tags that mapped to the chicken reference genome sequence [38] (ftp://ftp.ensembl.org/pub/release-101/fasta/gallus_gallus/dna/) were blasted with Bowtie software (v 2.1.0) using the known miRNA precursor/miRNA mature sequence as found in chicken miRBase (http://www.mirbase.org/). The unmatched tags were aligned to other RNA (exon, intron, repeat, GenBank database, Rfam database), identified and removed ribosomal RNA (rRNA), small cytoplasmic RNA (scRNA), small nucleolar RNA (snoRNA), small nuclear RNA (snRNA) and transfer RNA (tRNA), exon, intron [39], and novel miRNAs were predicted using miRDeep (v 2.0.0.5) [40,41]. To further analyze miRNAs expression levels in different groups, fold change values were calculated using the DEGSeq (v 1.18.0) and significant differential miRNAs were selected based on the threshold of |log2Fold| > 1, *p*-value < 0.05 [42].

### 2.4. miRNAs Target Gene Functional Annotation

Target genes prediction of DEMs was implemented using miRNA target gene prediction software (RNAhybrid, Miranda, Target Scan) and the outcomes were combined into a final result. To comprehensively describe the properties and function of target genes, GO terms [43] and KEGG pathway enrichment analysis [44] (http://www.genome.jp/kegg/) of differentially expressed genes (DEGs) were implemented using the GOseq (v 1.16.2), in which gene length bias was corrected. GO terms and KEGG pathways with a *p*-value < 0.05 were considered significantly enriched by DEGs.

### 2.5. Validation of DEMs and Target Gene

The candidate DEMs, miR-34a and miR-449a, miR-449c, were selected based on the deep sequencing array results. To explore the relationship between immune cytokines and miRNAs, potential target genes of DEMs were predicted using the two most popular databases, namely, Target Scan (http://www.targetscan.org/mamm_31/) and miRDB (http://mirdb.org/). From this, IL-2, IL-4, and IL-12α (Table 1) were predicted to be the target genes of miR-34a and miR-449c. Through the analysis of the Biodbnet database (https://biodbnet-abcc.ncifcrf.gov/db/db2db.php), it is found that the two target genes are enriched on the signal pathway of cytokine-cytokine receptor interaction, which is also the main enrichment pathway of other DEGs. Then two vectors containing target genes were constructed and cotransfected with miRNA mimics into 293T cells. Luciferase activity was measured using a dual-luciferase reporter assay kit (Promega, Madison, WI, USA). All of the experiments were performed in triplicate.

#### 2.5.1. Plasmid Construction

Plasmid psiCHECK2 contains the Renilla luciferase and Firefly luciferase genes, as well as two restriction sites, XhoI and NotI, located downstream of the Renilla luciferase gene. Firefly luciferase was used as an internal reference, and Renilla luciferase was used as an index to detect the expression of target genes. Gallus DNA was prepared for target gene amplification and recombinant construction, and the target gene primer containing NotI and XhoI sites were devised for amplification. Purified target genes for PCR production and psiCHECK2 were double digested with NotI and XhoI at 37 °C for 12 h. The PCR reaction mixture contained 0.1–0.5 µg of DNA, 1 µL of NotI, 2 µL of XhoI, 2 µL of 10× Buffer O, and ddH_2_O up to 50 µL. Purified enzyme-digested products and enzyme-digested psiCHECK2 were linked by T4 Ligases at 16 °C for 12 h (DNA 2.5 µL, psiCHECK2 2.5–4 µL, 10× Ligase Reaction Buffer 2 µL, T4 DNA Ligase 5 unit/1 µL, and ddH_2_O up to 20 µL). Subsequently, linked products were transformed into Escherichia coli. The screened monoclonal bacterium was checked by sequencing, and the plasmid including the target gene fragment was extracted.

#### 2.5.2. 293T Cell Culture and Transfection

The 293T cells (American Type Culture Collection, Manassas, VA, USA) were cultured in Dulbecco’s modified eagle medium (DMEM) (Gibco, Waltham, MA, USA) containing 10% (*v*/*v*) heat-inactivated fetal bovine serum (FBS) (Gibco) and 1% penicillin-streptomycin (Sigma, St. Louis, MO, USA) in a 5% CO_2_ incubator at 37 °C. According to the instructions of Lipofectamine 2000 reagent (ThermoFisher Scientific, Waltham, MA, USA), 0.2 µg of target gene expression plasmids or empty vector, and 5 pmol of miRNA mimic were cotransfected into 293T cells at 70–90% confluency in 96-well plates. After 6 h, removed the incubation solution and then continued to culture in 10% FBS.

#### 2.5.3. Dual-Luciferase Reporter Gene Assay

After transfection 24 h, according to the manufacturer’s instructions, luciferase reporter analyses were performed using a dual-luciferase reporter gene assay kit (Promega, Madison, WI, USA), and Firefly and Renilla luciferase activity were measured. The 293T cells were removed from growth media and lightly washed in 1× PBS. After the culture plate added 100 μL Passive Lysis Buffer (PLB) and then shaken at room temperature for 15 min, the solution was transferred to the PCR tube, centrifuged by 12,000 rpm for 5 min, and collected the supernatant. Add the collected supernatant at 5 μL per hole in the enzyme plate, then add 50 μL luciferase assay reagent II (LARII) to each well, gently shake and mix, and detect the fluorescence value of Firefly luciferase. Then 50 μL of Stop&Glo, was added to each well to detect the fluorescence value of Renilla luciferase. The relative Firefly luciferase activity was calculated by normalizing to Renilla luciferase activity. All experiments were performed in three independent biological replications, and reactions of each sample were carried out in triplicate. All data were expressed as means ± the SEM and determined by one-way ANOVA. *p* values < 0.05 were considered statistically significant.

### 2.6. RNA Isolation, cDNA Synthesis, and qRT-PCR Analysis

To further verify the regulatory relationship between miRNA and target genes, and its effect on other immune cytokines, we measured the expression levels of miRNA and its target genes as well as other immune cytokines by qRT-PCR. Three spleen tissue samples were used to isolate RNA using RNAiso Reagent (Takara Bio Inc. Otsu, Shiga, Japan). Total RNA purity and concentration were measured by Nanodrop 2000, and the integrity of RNA was checked by 1.5% agarose gel electrophoresis. The A260/A280 ratio was expected to range between 1.8 and 2.0. cDNA synthesis was performed using a two-step method offered by the PrimeScriptT II 1st Strand cDNA Synthesis Kit (Takara Bio Inc. Otsu, Shiga, Japan). Primers (Appendix A) were designed and produced by Sangon Biotech (Shanghai, China) and miRNA-specific stem-loop primers for cDNA synthesis of miRNA (Appendix A) were devised according to a previously described method [45,46]. The expression levels of immune-related target genes (IL-2, IL-12α, gga-miR-34a-5p, gga-miR-449c-5p) and other immune cytokines (IL-1α, IL-1β, IL-4, IL-6, IL-12β, and INF-γ) were measured with the SYBR Primer Ex Taq™ II kit(Takara Bio Inc. Otsu, Shiga, Japan) in LightCycle480 (Roche Molecular Biochemicals, Mannheim, BW, Germany). The qRT-PCR reaction contained SYBR Green 1 dye (5 μL), RT primer mix (1 μL), sample cDNA (1 μL), RNAase free water up to 10 μL. The program was started at 95 °C for 10 min followed by 45 cycles each of: 95 °C for 10 s, 60 °C for 10 s, 72 °C for 10 s, and 72 °C for 6 min. The results were normalized to the expression levels of 28s rRNA using the 2^−ΔΔCt^ method for quantification. All of the experiments were performed in triplicate.

### 2.7. Statistical Analysis

All experiments were performed in three independent biological replications, and reactions of each sample were carried out in triplicate. SPSS 26.0 (IBM Corporation, Armonk, NY, USA) was adopted for data statistical processing, all data were expressed as means ± the SEM and determined by one-way ANOVA analysis, *p* values < 0.05 were considered as statistically significant.

## 3. Results

### 3.1. Preliminary Analysis of the Raw Data

The average of the total raw reads was 15,337,421, 14,128,874, and 15,152,720 in the NS, CS, and HS groups, respectively (Table 2). After filtering, the total clean reads, mean values of 14,098,290 (91.92%), 13,428,578 (95.04%), and 14,230,205 (93.91%) were detected in the three groups NS, CS, and HS, respectively. As shown in Table 2, about 87% of the sequences were matched to the reference genome, the reads were matched to the miRbase and we predicted the novel miRNAs, the following numbers were found: we found 266 known miRNA and 80 novel miRNA for the NS group, 247 known miRNAs and 78 novel miRNAs in the CS group, and 252 known miRNAs and 83 novel miRNAs in the HS group.

DEMs responding to acute heat/cold stress were identified in the NS group, CS group, and HS group. According to |log2Fold| > 1, *p*-value < 0.05, there were 33 differentially expressed miRNAs in the NS-HS group, which included 17 significantly up-regulated miRNAs and 16 significantly downregulated miRNAs. Only 7 miRNAs expressed a significant level in the NS-CS group (six upregulated and one downregulated). The CS-HS group had 37 significantly expressed miRNAs which included 20 significantly upregulated miRNAs and 17 significantly downregulated miRNAs (Figure 1).

Furthermore, it was evident that five miRNAs, including miR-32-5p, miR-33-5p, miR-449a, miR-142-5p, and miR-34a-5p, showed a higher expression level in both the CS-HS group and NS-HS group (Appendix A). However, miR-6631-5p in both the CS-HS group and NS-HS group showed low expression levels. In CS-HS group (Appendix A), miR-15, miR-138-5p, miR-449c, miR-101-2-5p, and miR-103-2-5p were significantly upregulated, whereas miR-100-5p, miR-99a-5p, and gga-miR-1b-3p were significantly downregulated. To further explore differential expressed miRNAs, a heat map was plotted to demonstrate the miRNAs expression profile (Figure 2), which showed that miR-142, miR-34a, miR-449a, and miR-449c had the same expression pattern.

### 3.2. Gene Ontology (GO) Enrichment and Pathway Analysis

To investigate the biological function of target genes corresponding to the screened DEMs, these target genes were enriched in the GO database. Based on sequence alignment, target genes were distributed to each term and the results were assembled into a bar chart. The cellular component nuclear nucleosome is strongly correlated with differentially expressed miRNA target gene functions (Figure 3). However, the target genes of differently expressed miRNAs were not significantly enriched in molecular function. Many biological processes such as DNA methylation on cytosine, regulation of gene silencing, chromatin silencing at rDNA (Ribosomal DNA), and protein heterodimerization are remarkably related to miRNAs regulation. One of them, the cellular response to stress, was our focus in this research and differently expressed miRNAs exerted an important impact on this biological process.

To understand the regulation pattern of DEMs and their functions, pathway enrichment for CS-HS, NS-CS, and NS-HS groups was employed using the KEGG database. The significance of differential gene enrichment in each pathway entry was calculated using a hypergeometric distribution test. Cytokine-cytokine receptor interaction was significantly enriched in the HS group compared to the NS group (Figure 4). The neuroactive ligand-receptor interaction was significantly enriched in the CS group compared to the NS group (Figure 5). In the CS-HS group, the target genes of differently expressed miRNAs were significantly enriched in systemic lupus erythematosus and cytokine-cytokine receptor interaction pathway (Figure 6).

### 3.3. miRNAs Target Genes Identification

The target genes IL-2, IL4, IL-12α, were checked by Dual-Luciferase Reporter Assay. It was noted that the overexpression of gga-miR-34a and gga-miR-449c led to a significant decrease in IL-2 and IL-12α expression levels (Figure 7A,B) but did not influence IL-4 expression level compared to the negative control. This indicated that IL-2 and IL-12α were the target genes of gga-miR-34a and gga-miR-449c.

### 3.4. Quantitative Real-Time PCR (RT-qPCR) Analysis

To further explore IL-2 and IL-12α expression levels in spleen tissue, qRT-PCR was used to check whether IL-2 and IL-12α were the potential targets of gga-miR-34a-5p and gga-miR-449c-5p. In conclusion, gga-miR-34a-5p and gga-miR-449c-5p expression levels were consistent with our sequencing results, except that the gga-miR-34a-5p expression level in the HS group was higher than that in the CS group. The expression of two miRNAs in the HS and CS groups was significantly upregulated compared to that in the NS group (Figure 8). Additionally, the expression level of IL-2 in the HS group remarkably decreased compared to the NS group, and IL-12α expression level in the HS group displayed a strikingly decrease compared to the CS group. From the perspective of miRNAs and target genes expression level in the HS group, the existence of miR-34a and miR-449c prohibits the expression of IL-2 and IL-12α.

Furthermore, cytokines profile in splenic tissue was constructed to elucidate the immune mechanism in acute cold/heat stress. In comparison to the CS and NS group, IL-1β significantly up-regulated at extremely higher expression levels in the HS group (Figure 9), and the expression level of IL-4 in the HS group was significantly higher than that in the CS group. IL-6 expression levels showed an upward trend in the HS group compared to other groups, but the expression level of IL-1α showed a downward trend. The INF-γ expression level was higher in the HS and CS group than in the NS group.

## 4. Discussion

Through miRNA sequencing analysis, only 7 miRNA were found in the CS-NS group, however, we found 33 and 37 differential miRNA in the spleen tissues of the HS-NS group and HS-CS group, respectively. Compared with cold stress, heat stress is more likely to affect the expression of miRNA. Moreover, we determined that miR-34a and miR-449c were important factors regulating the interaction of immune cytokines. They belong to the same family and target IL-2 and IL-12α simultaneously.

As a component of the p53 pathway, miR-34a is widely expressed in immune cells, including B cells, T cells [47]. Studies have shown that the mir-34 family is a p53 target and a potential tumor suppressor for regulating processes such as proliferation, apoptosis, and metastasis [48,49], and is closely related to the cell progression and signal pathways of many diseases [50,51,52]. It has also been reported that miR-34a is expressed in the lungs of chickens infected with AIV, targeting 14 immune-related genes and four AIV genes [53]. Many studies have shown it to be an important regulator of human immune and inflammatory responses, as miR-34a is the main hub for regulating the immune system [54], and together with other miRNAs, it regulates the response process of T cells. In addition, as two members of the same family, gga-miR-449c and gga-miR-34a have similar functions and are highly conserved among different species. David et al. found that the sharp downregulation of sperm miR-449 and miR-34 family members and the severe stress was associated with a reduction in sperm quality and fertility in men [55]. Cao et al. found that the severe decrease in the expression of miR-449a and miR-34c in the early embryos of stressed male mice may alter brain development and spermatogenesis in more subtle ways than in gene knockout mice. At the same time, in patients with acute respiratory distress syndrome, upregulation of serum miR-34 and miR-449 may play a role in promoting airway terminal epithelial differentiation [56].

These data effectively show that miR-34a and miR-449c can regulate body immunity under the effect of stress. Although these studies are all based on humans, they can provide insight into poultry immunity, and studies in different animals have shown that miR-34a is indeed related to immune regulation, and that it is always up-regulated under heat stress. Yu et al. [35] found that Rno-miR-34a and Rno-miR-34b were significantly up-regulated in the small intestine of rats after heat treatment, suggesting that miR-34 may also be involved in heat stress-induced intestinal epithelial cell apoptosis. C2dat1 promotes cell proliferation, migration, and invasion in osteosarcoma cells by targeting miR-34a-5p [57], and in osteoarthritis (OA), lncRNA-TFC1 promotes the proliferation of chondrocytes by associated with miR-34a [58].

In addition, through the KEGG pathway enrichment analysis of the target genes of DEMs, it can be seen that the DEMs are mainly involved in the regulation of immune-related cytokine–cytokine receptor interaction. We verified that miR-34a and miR-449c could significantly reduce the expression of IL-2, IL-12α target genes. Consistent with many studies, miR-34a and miR-449c are closely related to cytokines. IL-1β induces the expression of miR-34a in OA [59]. The expression of miR-34a can directly or indirectly upregulate the expression of IL-6 and IL-8, leading to senescence, cell cycle migration, and self-renewal [60]. In addition, compared with that of wild type (WT) high-fat diet (HFD) mice, the high expression level of IL-10 was detected in white adipose tissue of epididymis of miR-34a-/-HFD mice. The levels of IL-5 and chemokine CXCL9 in miR-34a-/- mice fed a HFD were higher than those in the wild-type (WT) control group [61], indicating that miR-34a is positively correlated with IL-6 and IL-8, and negatively correlated with IL-10 and IL-5. Although no previous studies have directly shown that miR-34a and miR-449c have targeting effects on IL-2 and IL-12α, it can be seen from these results that miR-34a have a synergistic effect with miR-449c, and the immunomodulatory effect of miR-34a has been reported by many studies.

Heat stress affects the immune system by changing the T-helper 1 (Th1)/T-helper 2 (Th2) ratio [62]. Th1 cells activate cellular immunity and inflammatory responses such as IL-1β, IL-2, IL-6, IL-8, IL-12, IFN-γ, and tumor necrosis factor-α (TNF-α), whereas Th2 cells promote anti-inflammatory responses IL-4, IL-10, IGF-10, and IL-13 [63]. The ability of animals to selectively produce Th1 cytokines and Th2 cytokines is an important component in regulating the Th1:Th2 cytokine balance [60]. Heat stress may target IL-2 and IL-12 through miR-34a and miR-449c, thereby affecting the ratio of TH1/TH2.

IL-2 is a soluble leukocyte stimulating factor produced by T cells. The main function of IL-2 is as a crucial growth and expansion factor for T helper cells. It indirectly influences the production of virtually all T cell-derived cytokines and promotes the proliferation of both CD4(+) and CD8(+) T cells, and it may block autoimmunity by affecting the development of CD4(+), CD25(+) T-regulated cells during thymus development [64,65,66]. However, the overexpression of miR-34a in CD8(+) T cells reduces the killing ability of T cells [67]. miRNA expression analysis of different blood cell types showed that miR-34a was overexpressed in CD3(+) T cells [68]. Such a conclusion is consistent with our results. IL-12 is a natural killer (NK) cell-stimulating factor with multiple biological effects on peripheral blood lymphocytes, and the synergistic effect of IL-12 and IL-2 enhanced the cytotoxic activity of NK [69]. As one of the main controllers of the immune response, many pro-inflammatory cytokines such as IL-6, IL-1β, and TNF-α have been reported to respond to stress and immune disorders, and IL-6-related pathways also play a key role in regulating acute heat stress [70].

The cytokine spectrum showed that the expression of IL-1β in the HS group was significantly higher than that in the NS and CS groups. The expression level of IL-6 showed a moderate upward trend. At the molecular level, pro-inflammatory cytokines such as IL-1 and TNF-α have important effects on disease behavior [71], including the decrease in performance parameters caused by acute heat stress [13]. It is suggested that IL-1β plays an important role in acute heat stress. Related studies have shown that IL-1β treatment can significantly upregulate the expression of miR-34a in human articular chondrocytes [72]. Similarly, miR-21, miR34a, and miR-146a increased significantly through the induction of IL-1β and TNF-α in mouse insulinoma 6 (MIN6) cells [73]. qRT-PCR results showed that the expression level of miR-34a in the HS group was higher than that in the NS group, and it is suggested that the high expression of IL-1β in the acute HS group may promote the expression of miR-34a. The number of spleen lymphocytes was determined by flow cytometry. In response to miR-34 antagonists, the Th1 producing IFN-γ and the Th17 cell population producing IL-17α decreased significantly [31]. Traditional dendritic cells overexpressed by miR-34a also produce large amounts of IL-17α and inhibit T cell activation [74], and similar to our results, the expression levels of INF- γ, and miR-34a in the HS group and CS group were higher than those in the NS group. It was shown that miR-34a could also promote the expression of INF-γ and IL-17α, and the promotion of IL-17α was confirmed by stimulating the activity of NF-κB and inducing fibroblasts to secrete IL-6 [75]. After stimulating valve interstitial cells (VICs) with IL-6, the expression of miR-449c-5p decreased significantly over time [76]. As an inflammatory mediator, IL-6 participates in the regulation of acute inflammatory responses. In HepaRG cells, IL-17 (also known as IL-17α) enhances the stability of IL-6 mRNA, increasing the level of IL-6 protein [77]. Our results also showed that the expression of IL-6 in the acute HS group was higher than that of other groups, which indirectly confirmed that IL-17α activated the expression of IL-6. In short, IL-1β significantly up-regulated and induced the expression of miR-34a during acute heat stress. The increased expression of miR-34a and miR-449c directly down-regulated the expression of IL-2 and IL-12α. At the same time, the up-regulation of miR-34a could induce a high expression level of INF- γ, and IL-17A. Then IL-17α promoted the increase of IL-6 expression.

## 5. Conclusions

IL-2 and IL-12α are target genes of miR-34a and miR-449c, the overexpression of gga-miR-34a and gga-miR-449c led to a significant decrease in IL-2 and IL-12α expression levels. In addition, IL-1β plays an important role in the regulation of the response to acute heat stress in broilers by inducing the upregulation of miR-34a, and upregulation of miR-34α promotes the differentiation of INF-γ produced Th1 cell differentiation. Moreover, the overexpression of miR-34a also promotes the expression of IL-17 and thus promoting the expression of IL-6. Our research provides a theoretical basis for heat/cold resistance breeding of broilers, but further study is needed on the regulation of immune cytokines affected by miRNAs, that ultimately affect heat stress.

## Figures and Tables

**Figure 1 animals-10-02168-f001:**
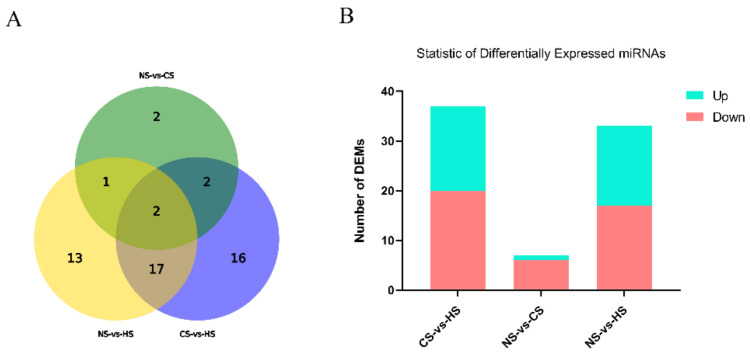
(**A**): Statistics of overlap between differentially expressed miRNAs (DEMs) groups: The circles represented by different elements represent different conditions, and the numbers represent the number of different microRNAs (miRNAs); (**B**): Statistics of DEMs among different groups: The abscissa represents the comparison between samples, and the ordinate represents the number of significantly DEMs. Red represents up-regulated miRNA, and blue represents down-regulated miRNA.

**Figure 2 animals-10-02168-f002:**
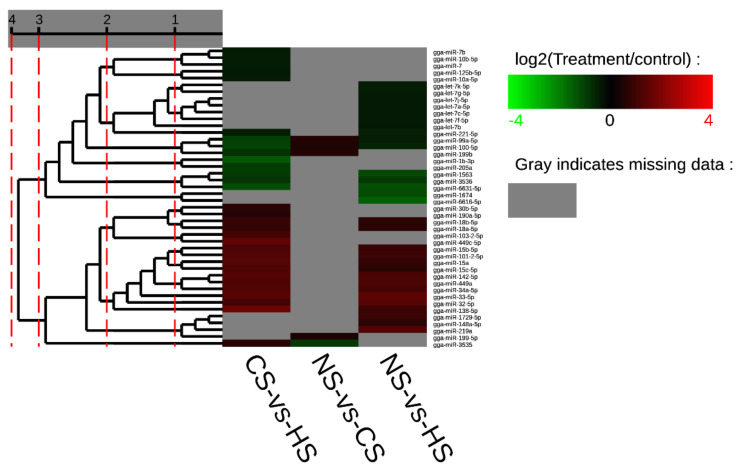
Cluster map of differential expression miRNA. Each column represents an experimental condition, and each row represents a gene. Colors represent expression level (logarithm). The redder the color, the higher the expression of miRNA, vice versa for the blue color. Gray indicates that the miRNA does not exist in the corresponding sample.

**Figure 3 animals-10-02168-f003:**
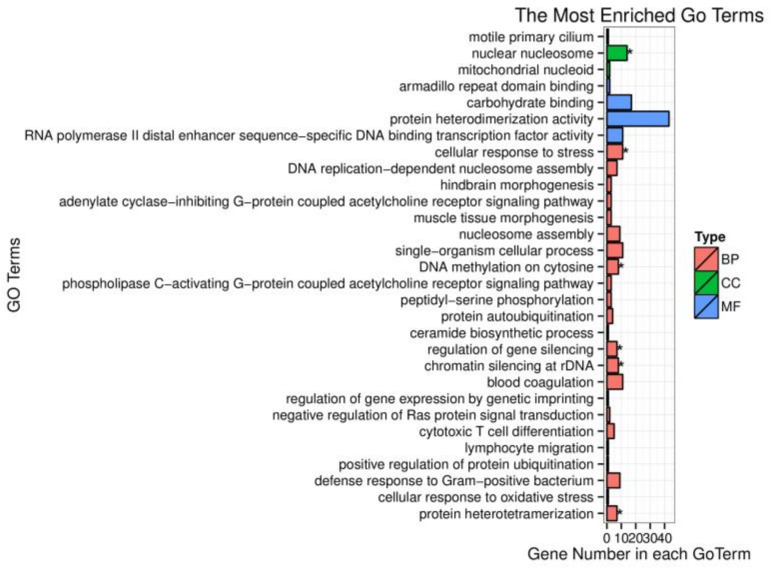
Gene ontology (GO) enrichment analysis of differentially expressed miRNA target genes. BP represents biological process; CC represents cellular component; MF represents molecular function; the symbol * means *p*-value < 0.05.

**Figure 4 animals-10-02168-f004:**
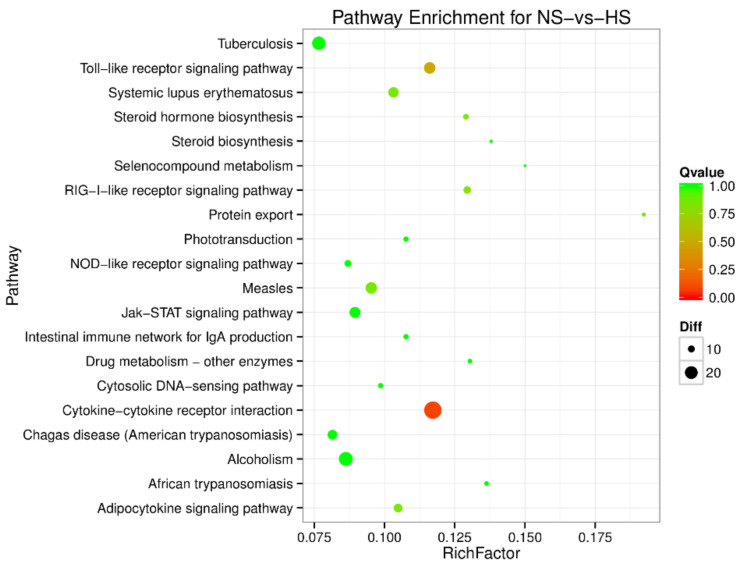
Pathway enrichment analysis of differentially expressed miRNA target genes for normal (NS)-vs-heat stress (HS). The larger circle area means the more differently expressed gene number. The Q value represents a significant level (*p*-value < 0.05).

**Figure 5 animals-10-02168-f005:**
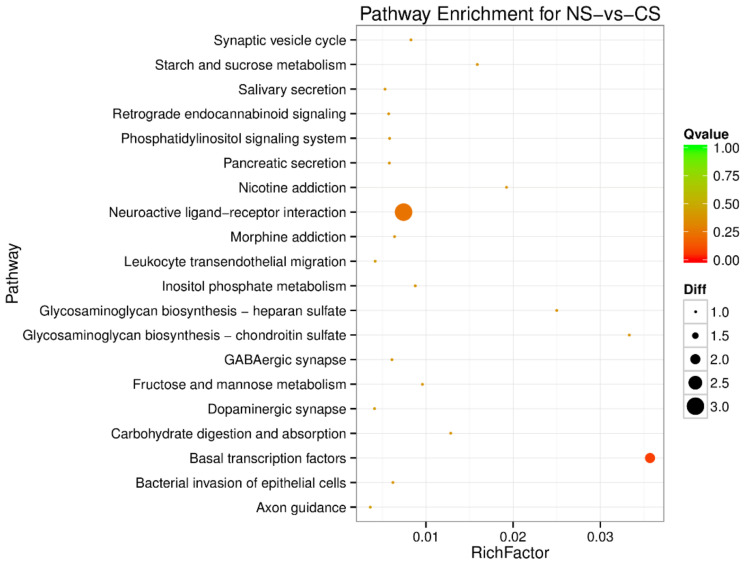
Pathway enrichment analysis of differentially expressed miRNA target genes for NS-vs-cold stress (CS). The larger circle area means the more differently expressed gene number. The Q value represents a significant level (*p*-value < 0.05).

**Figure 6 animals-10-02168-f006:**
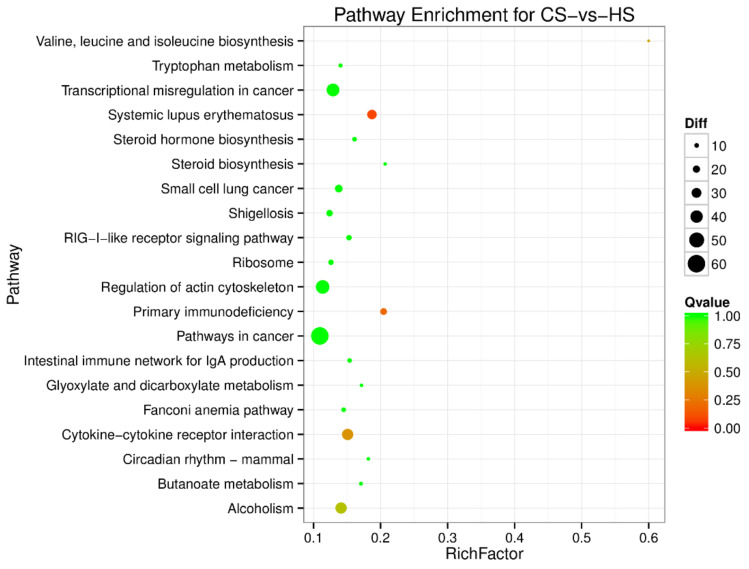
Pathway enrichment analysis of differentially expressed miRNA target genes for CS-vs-HS. The larger circle area means the more differently expressed gene number. The Q value represents a significant level (*p*-value < 0.05).

**Figure 7 animals-10-02168-f007:**
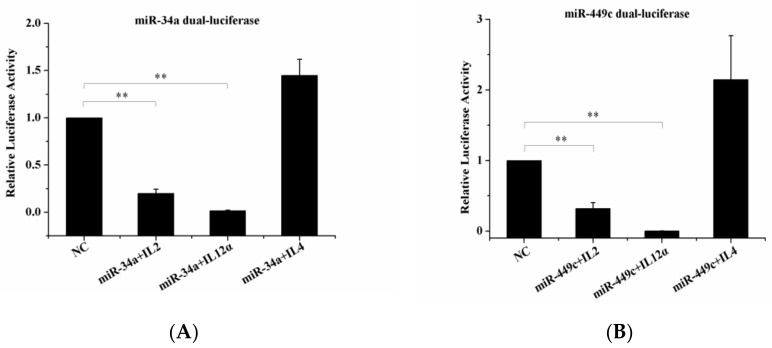
Dual-luciferase reporter gene assay confirmed targeted binding between IL-2, IL-12α, and miR-34a, miR-449c. (**A**): 293-T cells were transfected with miR-449c mimic or an NC vector and then analyzed for miR-34a expression. (**B**): 293-T cells were transfected with miR-34c mimic or an NC vector and then analyzed for miR-449c expression. ** *p* < 0.05 compared with the NC group.

**Figure 8 animals-10-02168-f008:**
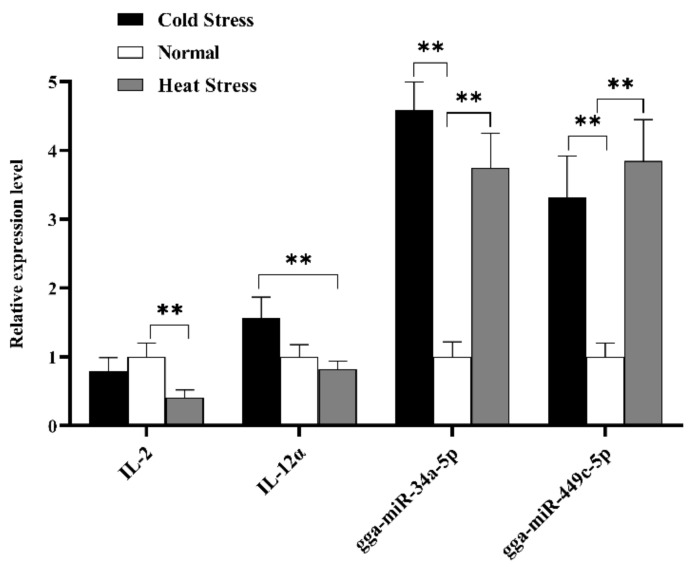
Expression analysis of selected two DEMs and two target genes in splenic tissue of different broiler (HS, NS, CS) by RT-qPCR methods. The *x*-axis represents different miRNA, and the *y*-axis represents the relative expression level of miRNAs or mRNA targets. Data are presented as mean ± standard error. * on the top of lines or bars indicated a significant difference (** *p* < 0.05) between heat stressed and normal condition.

**Figure 9 animals-10-02168-f009:**
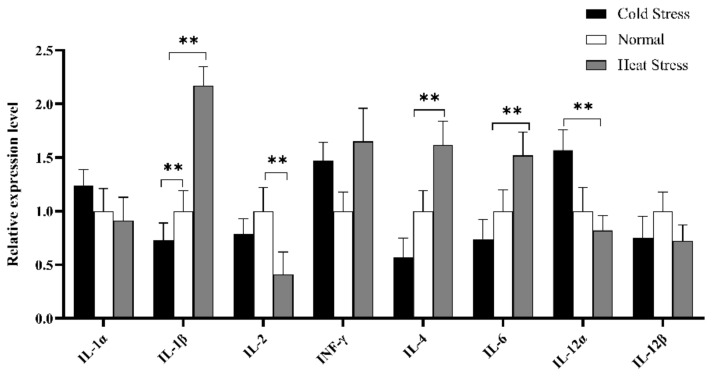
Expression analysis of immune cytokines profiles in splenic tissue of different groups (HS, NS, CS) broiler by RT-qPCR methods. The *x*-axis represents different cytokines, and the *y*-axis represents the relative expression level of cytokines. Data are presented as mean ± standard error. * on the top of lines or bars indicated a significant difference (** *p* < 0.05) between heat stressed and normal condition.

**Table 1 animals-10-02168-t001:** The selected target genes of miRNAs.

microRNA ID	Gene ID	Description
gga-miR-34a-5p	373958	IL2//interleukin 2
gga-miR-34a-5p	416330	IL4//interleukin 4
gga-miR-449c-5p	416330	IL4//interleukin 4
gga-miR-34a-5p	407090	IL12α//interleukin 12α
gga-miR-449c-5p	407090	IL12α//interleukin 12α
gga-miR-449c-5p	373958	IL2//interleukin 2

The target gene comes from software (Target Scan and miRDB) prediction.

**Table 2 animals-10-02168-t002:** Statistics of small RNA sequences from broiler’s spleens.

Type	Normal	Cold Stress	Heat Stress
Total Raw Reads	15,337,421	14,128,874	15,152,720
Total Clean Reads	14,098,290	13,428,578	14,230,205
Total Clean Reads Ratio (%)	91.92	95.04	93.91
Total Invalid Adapter Reads	243,626	204,855	170,239
Total Invalid Adapter Reads Ratio (%)	1.60	1.40	1.12
The Total Alignment Rate with The Reference Genome (%)	88.85	86.51	86.88
Detected Known miRNA Number	266	247	252
Detected Novel miRNA Number	80	78	83

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
