# Peer review of "The Mediation of miR-34a/miR-449c for Immune Cytokines in Acute Cold/Heat-Stressed Broiler Chicken"

_animals, 2020, doi:10.3390/ani10112168_

Round 1
Reviewer 1 Report
Review: The mediation of miR-34a / miR-449c for immune cytokines in acute cold/ heat-stressed broiler chicken R1
The authors have much improved the manuscript. There are just a few textual thinks that could still be revised:
“after matched to the miRbase and predicted the novel miRNAs, the NS” --> “. The reads were matched to the miRbase and we predicted the novel miRNAs, the following numbers were found: we found 266 known miRNA and 80 novel miRNA for the NS group, 247 known miRNAs and 78 novel miRNAs in the CS group, and 252 known miRNAs and 83 novel miRNAs in the HS group”
Figure 1: increase the font size of the labels a bit for readability.
“research. and differently expressed miRNAs exerted an important impact on this biological process” --> “research and differently expressed miRNAs exerted an important impact on this biological process”
“In the CS-HS group, the target genes of differently expressed miRNAs significantly enriched in systemic lupus erythematosus and cytokine-cytokine receptor interaction pathway” --> “In the CS-HS group, the target genes of differently expressed miRNAs were significantly enriched in systemic lupus erythematosus, and cytokine-cytokine receptor interaction pathway”
“Li et al. [43]studied the expression” à space after citation
“Our research provides a theoretical basis for resistance breeding of broilers, but further study is needed on the regulation of immune cytokines by more miRNA affected by heat stress” à
“Our research provides a theoretical basis for heat/cold resistance breeding of broilers, but further study is needed on the regulation of immune cytokines affected by miRNAs, that ultimately affect heat stress”
Reviewer 2 Report
For overall experiments, n=3 is too low to draw sound conclusions. For example; in figure 8 IL-12alpha is significantly different between heat and cold stress conditions. However, in figure 9 IL-12alpha is not significantly different between heat and cold stress conditions. There is a similar issue with IL-2 expression in different conditions on figures 8 and 9.
Authors should consider using flow cytometry to examine cytokine profile in the spleen.
Can the authors provide direct evidence which shows that miR-34a, miR-449c bind and cleave their target genes?
Reviewer 3 Report
The authors have revised well.
Author Response
Dear Reviewer:
On behalf of my co-authors, we thank you very much for giving us an opportunity to revise our manuscript. We appreciate the previous constructive comments and suggestions on our manuscript entitled “The mediation of miR-34a / miR-449c for immune cytokines in acute cold/ heat-stressed broiler chicken” (ID: animals-995273). Those comments are all valuable and very helpful for revising and improving our paper, as well as the important guiding significance to our researches.
Special thanks to you for your good comments.
Yours sincerely,
Tao Li.
Reviewer 4 Report
Dear Authors,
the revisions you made improved manuscript’s quality, anyway you need to furtherly work on it.
English is quite often coarse and editing inaccurate, please totally revise written English throughout the manuscript.
Pay attention to editing, sentences, paragraphs and spaces between words.
Simply summary has been improved, revise the last sentences when listing your actions.
Abstract has been improved, pay attention to experimental groups description (T, RH).
Introduction has been improved, some sentences of the first part of the results section should be moved here because not strictly related to your results.
Materials and methods have been improved, anyway, some mistakes are still present in abbreviations and groups’ descriptions. Try to use animal related protocol (poultry when present) as references.
Results have been improved, in tables and pictures try to adequate your descriptions to Animals style and reference papers examples.
Discussion has been slightly improved, move all the bibliographical observations not related to your results to the introduction section. Report and comment your results, they are interesting to be commented with the poultry related and other references you reported. you did well in the last part of the chapter.
Conclusions the sentence you added is not enough to improve the section which is still very poor in content. You have interesting results: sum the up in this section.
Author Response
Please see the attachment!

This manuscript is a resubmission of an earlier submission. The following is a list of the peer review reports and author responses from that submission.
Round 1
Reviewer 1 Report
Review: The mediation of miR-34a / miR-449c for immune cytokines in acute cold/ heat-stressed broiler chicken
The authors present a research article about differentially regulated miRNAs under cold and heat stress affecting immune cytokines. The study is well-designed, and the results are presented accordingly. However, before publication, some substantial improvements needs to be done. Unfortunately there are no page numbers, so I directly copied the text I am referring to.
Comments:
Simple summary and Abstract
Why were miR-34a and miR-449c chosen as the research objects. Were these the most significantly differentially expressed miRNAs?
“and the clean reads filtered from the raw reads” change to “after cleaning the filtered reads”
What do you mean with different databases? I assume the reads are first mapped to the chicken reference genome? And then further downstream analysis is performed using different databases.
“miR-34a and miR-449a, miR-449c” à miR-34a, miR-449a, and miR-449c
“in acute cold/ stress”: Why the “/”? should it be: “in acute cold/heat stress”?
“Our found have” à “Our finds have”
Introduction
“do not have a sweat gland”: Not proper English, please rephrase.
“the pathogen such as” à “pathogens such as”: should be plural
MicroRNA (miRNAs) is the small (18-25nt) non-coding RNA à I think ‘is’ is more appropriate here instead of ‘the’.
“regulation of the immune” à “regulation of the immune system”
Remove “as we all know”
“the concrete mechanism of it is still not unclear.” à remove “not” in this sentence.
Start the last paragraph of this study with: “The aim of this study is …”
Meanwhile, we try to find some miRNAs target genes related to immune à Meanwhile, we try to find some miRNAs target genes related to the immune system.
miRNAs exert significant à miRNAs that exert significant
Material & Methods
reviewed and was approved à was reviewed and approved
“A total of 30 Rose 308 broiler chickens”: What is 30 Rose 308 broiler???
and predict the novel miRNAs using miRDeep à and novel miRNAs were predicted using miRDeep.
I don’t understand this sentence: “The clean tags were aligned to different databases(GenBank, Rfam, repeat), identified rRNA, tRNA, snRNA, snoRNA, exon, intron”. How and where was it aligned to? So you looked at different genes, RNAs etc.. Please be much more specific here.
Please rephrase following sentence: “which it is also the main enrichment pathway of other DEGs”.
Results
3.1 chicken reference genome citation is missing.
“After total clean reads matched to the reference genome, the Normal group found 266 known miRNA and 80 novel miRNA. 247 known miRNAs and 78 novel miRNAs were found in the Cold stress group. As well, 252 known miRNAs and 83 novel miRNAs were checked in the Heat stress group.”
We aligned the cleaned reads to the reference genome after which we found 266 known miRNA and 80 novel miRNA (NS). 247 known miRNAs and 78 novel miRNAs in the Cold stress group and 252 known miRNAs and 83 novel miRNAs in the Heat stress group.”
The English can be improved throughout the manuscript, please make sure you perform proofreading by a native English speaker.
What was the total alignment rate of the miRNAs? I cannot find this anywhere?
“There is only 7 miRNA expressed a significant level in the N-C group. 37 significant expressed miRNAs were found in the C-H group”
You use “is” and “were” in one sentence, please keep this grammatically consistent (again, please do proofreading for the English).
Reference is missing at the end of this paragraph (before Figure 1).
Please check carefully the references to tables and figures, there are a lot of errors in the manuscript.
Strikingly upregulated à were strikingly upregulated
Missing “were” again before significantly downregulated.
“However, for the target genes of differently expressed miRNAs were not significantly enriched in molecular function” à remove “the” in this sentence.
“To knowledge about the regulation pattern of differently expressed miRNAs and how it works, pathway enrichment for CS-HS, NS-CS and NS-HS groups were employed using the KEGG database.” à this is not proper English
Discussion
“In 2014, 52 miRNAs were found in the serum of Holstein cows changed significantly due to heat stress, and these miRNAs regulated stress and immune response genes [34].” à English rephrase.
“Compared with cold stress, heat stress is more likely to affect the expression of miRNA.” . Is this statement based on your own observations that you find more DE miRNAs in heat stress group? Or is there also other papers that confirm this?
as a p53 target à is a p53 target
It is showed that miR-34a could also promote the expression of INF-γ and IL-17α to some extent à it is shown that…
Conclusions
I miss a concluding sentence indicating the overall impact and relevance of the study in the field. Please ass such a sentence (or two).
Reviewer 2 Report
In this study, authors aimed to study the effects of heat and cold stress on miRNA expression patterns in spleen in broiler breeder chicken.
Some parts of the results section can be moved methods.
Example; “Raw data were obtained from three libraries constructed from the NS group, CS group and HS group following the method of three in each group. The average of total raw reads was 15337421, 14128874 and 15152720 in NS group, CS group and HS group, respectively. After filtering, the total clean reads mean values with 14098290 (91.92%), 13428578 (95.04%) and 14230205 (93.91%) in three groups were matched to the chicken reference genome (Table 4Error! Reference source not found.). The total invalid adapter reads 243626 (1.60%), 204855 (1.40%) and 170239 (1.12%) were detected in three groups. After total clean reads matched to the reference genome, the Normal group found 266 known miRNA and 80 novel miRNA. 247 known miRNAs and 78 novel miRNAs were found in the Cold stress group. As well, 252 known miRNAs and 83 novel miRNAs were checked in the Heat stress group.”
References are not present in some sections of the manuscripts. Please read thoroughly before submission.
“3 spleen samples were selected randomly from each group for RNA isolation”
N number should have been higher for miRNA-seq. Higher n number is essential for Gene Ontology (GO) enrichment and Pathway analysis.
Figure legends must contain detailed information about experiments. Some figures don't have legends. It is very challenging to understand the experiments performed without technical details in the figure legends.
Y-axis doesn’t have title on some of the figures.
Reviewer 3 Report
I have reviewed the manuscript entitled “The mediation of miR-34a/miR-449c for immune cytokines in acute cold/heat-stressed broiler chicken” for a publication in Animals as an original article. The authors investigated miRNA-seq and qRT-PCR expression analyses for miRNA and target genes using spleen tissues. Analyses and results look reasonable. However, I think there are some questioned points. Therefore, I recommend this paper needs to be revised before publishing. My comments will be shown below:
Major comments
- The authors reared 30 chickens (n = 10 in each group). However, the authors used only n = 3 in each group. How about the authors use the remaining n = 7 for the validation step? If not, the number of animals should be 9 chickens (n = 3 in each group).
- In the Methods 2.5, how many samples were used for the validation?
- In the Methods 2.6, did the authors use the same samples (n = 3 in each group)?
- Did the authors investigate using ELISA for seeing cytokine profile? If not, the title of Figure 9 should be reworded. “Expression” should be included.
- Please add 2.7 Statistical analysis. The current version mentioned about statistics only in 2.5.3. It will be better that “all data” mean clearly the all data of the paper not only in luciferase assay in 2.5.3.
- Explanation of Figures should be included in Figures 7, 8, and 9. Like Figure 3, the authors should describe what asterisk means.
- In Figures 4, 5, and 6, Q values were shown. Did the authors set the threshold as significance? Please indicate it in Materials and Methods.
Minor comments
- What is the term “Error! Reference source not found.”? Please delete them.
- The authors should use “and” following the rule such as “A, B, C, D, and E” throughout the manuscript.
- In the Methods 2.5, “Cytokine-cytokine receptor interaction, which it is also” should be “Cytokine-cytokine receptor interaction, which is also”.
Reviewer 4 Report
Dear Authors,
please consider these suggestions
First of all consider these two papers:
Wei H, Zhang R, Su Y, et al. Effects of Acute Cold Stress After Long-Term Cold Stimulation on Antioxidant Status, Heat Shock Proteins, Inflammation and Immune Cytokines in Broiler Heart. Front Physiol. 2018;9:1589. Published 2018 Nov 13. doi:10.3389/fphys.2018.01589
Saleh KMM, Al-Zghoul MB. Effect of Acute Heat Stress on the mRNA Levels of Cytokines in Broiler Chickens Subjected to Embryonic Thermal Manipulation. Animals (Basel). 2019;9(8):499. Published 2019 Jul 29. doi:10.3390/ani9080499
Title:
You have different RH in your experimental groups: consider it in the title.
Simple Summary:
Correct “Simply in Simple”
Revise and rephrase: “The simple summary consists of no more than 200 words in one paragraph and contains a clear statement of the problem addressed, the aims and objectives, pertinent results, conclusions from the study and how they will be valuable to society. This should be written for a lay audience, i.e., no technical terms without explanations.”.
Abstract:
English language must be revised.
Experimental design and “materials and methods” description needs to be improved, animals and treatments must be described before results and conclusion.
Consider the specific sections corrections.
Introduction:
English language and grammar in particular (verbs, past tenses etc.) must be revised. Too many determinative articles throughout the papers. Focus on cold/heat stress and miRNA in poultry. Include adequate specific references.
Matherials and Methods:
2.1 Revise English language, abbreviation description and repetition, improve animals management procedures, add references about the treatments, stress duration, etc..
Tissue preparation: references
2.2 Add references
2.3 Editing, References
2.4 Editing, References, table 1 references.
2.5 Editing, References, Producing Companies descriptions.
Results:
3.1 tab 4?, add references, fig 1 revise and editing, fig 2 improve.
3.2 english language, references, move observations to discussion section, fig 3 improve, database addresses, add references, fig 4 editing.
3.3 revise English language.
3.4 add references
In general improve the description of your findings.
Discussion:
Move all the sentences not related to your results in the introduction section. Improve the discussion of your results
Conclusions:
Improve and rephrase focusing on your results, their effectiveness and their applicability in poultry industry.
